# *Toxoplasma gondii* in Foods: Prevalence, Control, and Safety

**DOI:** 10.3390/foods11162542

**Published:** 2022-08-22

**Authors:** Pablo-Jesús Marín-García, Nuria Planas, Lola Llobat

**Affiliations:** Department of Animal Production and Health, Veterinary Public Health and Food Sciences and Technology (PASAPTA), Facultad de Veterinaria, Universidad Cardenal Herrera-CEU, CEU Universities, 46115 Valencia, Spain

**Keywords:** toxoplasmosis, *Toxoplasma gondii*, control, food, detection

## Abstract

*Toxoplasma gondii* is an obligate intracellular parasite that causes toxoplasmosis, with approximately one third of the population around the world seropositive. The consumption of contaminated food is the main source of infection. These include meat products with *T. gondii* tissue cysts, and dairy products with tachyzoites. Recently, contamination has been detected in fresh products with oocysts and marine products. Despite the great health problems that are caused by *T. gondii*, currently there are no standardized methods for its detection in the food industry. In this review, we analyze the current detection methods, the prevalence of *T. gondii* in different food products, and the control measures. The main detection methods are bioassays, cell culture, molecular and microscopic techniques, and serological methods, but some of these do not have applicability in the food industry. As a result, emerging techniques are being developed that are aimed at the detection of multiple parasites simultaneously that would make their application more efficient in the industry. Since the prevalence of this parasite is high in many products (meat and milk, marine products, and vegetables), it is necessary to standardize detection methods, as well as implement control measures.

## 1. Introduction

Toxoplasmosis is a zoonotic disease that is caused by the obligate intracellular parasitic *Toxoplasma gondii*. This protozoon of the Apicomplexa phyla presents only felines as the definitive host, being the ones where the parasite can complete its life cycle. However, all warm-blooded animals, including mammals and birds, can act as intermediate hosts (Figure 1). In most hosts, *T. gondii* causes a lifelong latent infection in tissues such as skeletal and heart muscle, and the central nervous system, causing the disease. In humans, infection by *T. gondii* is particularly important in pregnant women and immunocompromized people. During pregnancy, the risk of fetal infection increases with gestational stage, increasing as gestation progresses [1]. Neonatal manifestations include hydrocephalus, microcephalus, intracranial calcifications, chorioretinitis, cataracts, convulsions, nystagmus, jaundice, petechiae, anemia, enlarged liver and spleen, prematurity, and severe intrauterine growth restriction [2,3]. Ocular manifestations also appear as chorioretinitis and retinal lesions [4]. In immunocompromized people, the neurological symptoms, such as encephalopathy, meningoencephalitis, cerebral mass lesions, headache, confusion, poor coordination, and seizures are usual [5], with toxoplasmic encephalitis being the most frequent manifestation in HIV patients [6], whereas the disseminated toxoplasmosis is more characteristic of transplant patients [7]. However, not only pregnant women and immunocompromized people may suffer the symptoms of *Toxoplasma* infection. Immunocompetent individuals can develop acute, chronic, and ocular toxoplasmosis. The acute toxoplasmosis is asymptomatic around 80% of individuals [8], and the symptoms in the other 20% includes fever, mononucleosis-like symptoms, with cervical posterior adenopathy, myalgia, and asthenia [9]. Although these symptoms are not relatively serious, the severity of infection depends on genotype of the parasite strain. In fact, infections with a highly virulent strain can produce fatal pneumonitis, myocarditis, meningo-encephalitis, and polymyositis [6]. In chronic toxoplasmosis, tachyzoites form bradyzoite cysts intraneuronal which are controlled but not eliminated by the immune system [10]. The immune response in the brain of patients produces brain inflammation, ventricular dilatation, disrupting neuronal structure and connectivity [11,12]. Although the symptoms of chronic toxoplasmosis have not been unraveled, several studies correlated these manifestations with neuropathies [13,14]. Related to ocular toxoplasmosis, it is the primary cause of infectious uveitis, presenting with retinochoroiditis [15].

*T. gondii* has a worldwide geographic distribution and an estimated 30% of the population is seropositive [16]. The genetic diversity of *T. gondii* around the world has been elevated, so more than 36 genotypes have been found [17]. The transmission of this parasite in humans may result from the ingestion of tissue cysts in raw or undercooked meat of infected animals, ingestion of raw vegetables, water that is contaminated with *T. gondii* oocysts from cat feces, and by vertical or transplacental transmission [18]. Although, the main route of infection in humans is through ingestion of contaminated food. In fact, it has been described that up to 50% of infections are caused by food transmission using a novel multiplex Polymerase Chain reaction (PCR) assay [19]. A study that was undertaken in school dining rooms of Colombia showed the presence of *T. gondii* in meat, water, cucumber, and guava juice, both inert and living surfaces [20]. In the last years, the concern about this zoonosis and its transmission has been increasing. In 2018, the EFSA recommended a serological screening of livestock to identify positive farms [21]. In the following year, the EFSA report found that food-borne transmission accounts for 40–60% of *T. gondii* infections [22]. The last report indicated positive samples of meat, fish, raw mollusks and shellfish, honey, and potable water, and *Toxoplasma* was included in category III of zoonotic agents to monitor, along with *Campylobacter* or *Yersinia* [23].

However, and despite the great health public problem that it poses, there are currently no specific detection criteria for *T. gondii* in food, and there are no standardized methods or validation procedures for its detection in the food industry. In fact, different direct and indirect detection techniques exist. Cat and mouse bioassays are the reference direct techniques to analyze the viability of the parasite, but these test are not commonly used due to the long time that is taken to obtain results, ethical issues, and great costs [18]. The alternative method are cell cultures which are limited in use because of the variability of the results depending the sample [24]. Other serological methods (indirect detection) have been developed such as immunofluorescent assay (IFAT), enzyme-linked immunosorbent assay (ELISA), latex agglutination tests (LAT), modified agglutination test (MAT), and more recently, a luciferase-linked antibody capture assay (LACA) [23,25]. The latest studies of *T. gondii* detection in food products have used serological techniques to improve the sensibility of these serological tests using different approaches. For example, Suwan et al. (2022) used a recombinant dense granule antigen 7 protein for the detection of parasites in blood samples [26]. In addition to these serological methods, other molecular techniques have been tested. Some protocols of PCR have been described as nested PCR, real-time PCR, loop-mediated isothermal amplification (LAMP), and others. However, the more sensitive and specific diagnostic tools to detect *T. gondii* are necessary [27], and the studies about their sensitivity and to unify the detection in different food products are essential to control of parasite infection by food consumption. The aim of this review is to delve into the current context of *T. gondii* infection through food, prevalence of different food products, its detection and control, and future perspectives.

## 2. Methods for *T. gondii* Detection in Food Products

Although *T. gondii* is a high priority foodborne zoonotic pathogen around the world, it is not systematically controlled [28]. At present, there are no specific regulations or ISO standards for the detection of *T. gondii* in any food matrix [21]. Even so, different methods are available to detect tachyzoites, tissue cysts, and oocysts in food products, including immunological and microscopical methods. These methods have an isolate and concentration stage, later applying direct detection methods to the sample. Molecular assays are used to detect the presence of *T. gondii* DNA in samples, while information on the viability and infectivity can be obtained by in vivo assays (usually in mice) or by in vitro culture techniques. A summary of these methods with sensitivity and type of food product where these methods have been used are shown in Table 1.

### 2.1. Animal Model Bioassay

This method allows the study of the infectivity of the oocysts and tissue bradyzoites of the parasite. For *T. gondii* detection, the cat bioassay works best, followed by the mouse bioassay [31]. In cats, the animals are fed with the test meat or tissue (up to 500 g) to analyze the presence of tissue cysts. A total of three weeks after exposure, the cat feces are tested for the presence of *T. gondii* oocysts, and serum samples may be analyzed to detect specific antibodies against the parasite [54]. The cat bioassay allows the detection of all stages, as tachyzoite, bradyzoite, and oocysts [55]. However, the bioassay in cats is carried out in few laboratories since it is an extremely expensive method and, in addition, the use of animals raises ethical problems [31].

Therefore, mice are the main animal model to evaluate the infectivity of oocysts. In this technique, 50 to 200 g of tissue are digested with acid pepsin or trypsin and a fraction of the sediment is inoculated in mice, generally intraperitoneally or subcutaneous; although mice can also be infected orally with *T. gondii* oocysts [56]. Typically, two to five mice are used per sample, monitored clinically, and when the mouse dies or is euthanized, brain or peritoneal fluid samples are analyzed for the presence of *T. gondii* by microscopy or PCR in addition to detect specific antibodies in serum. Immunosuppressive drugs can be administered to mice to increase the sensibility of the bioassay [34]. The sample size that can be tested is smaller than that which is used for the cat bioassay, as only a fraction of the digest is inoculated. Mouse bioassays are generally less expensive than cat bioassays, but they also present ethical issues [55]. These models have been used to evaluate the presence of infectious oocysts in water and shellfish samples [33], and to evaluate the impact of storage time and temperature on oocyst infectivity in raspberries and blueberries [57]. Bioassays are not useful for previously frozen samples as these tests are based on the viability of the parasite so it is not feasible for large-scale screening, and it does not quantify the intensity of infection [58,59]. However, for other types of samples, this method is still one of the most useful for the detection of viable parasites.

### 2.2. Cell Culture

Despite the fact that molecular techniques are very specific and sensitive, they only detect parasite DNA, regardless of whether it is viable. A solution to this problem is the detection of *T. gondii* by isolation in cell culture. The test sample is brought into contact with culture of different cell lines. If the parasite is present in the sample and it is viable, the culture cells will become infected, causing the tachyzoites to multiply, which can be observed with an inverted microscope after 3–10 days [60]. Cell cultures can be used as an alternative to bioassays in animals since the cost is lower and solves the ethical problem that is posed by bioassays [61]. Even so, it should be taken into account that cell cultures require perfect observation of the samples to avoid contamination and that they are less sensitive than the bioassay for detecting the parasite viability [62]. Moreover, it must be considered that to detect hazards in food or food outbreaks, faster results must be obtained so that contaminated food can be recalled. Another possibility would be the diagnostic use of methods based on tissue culture, although this is limited. Artificially digested meal or sediment homogenates have been tested with varying success rates. In a study with milk samples from different species of cattle based on tissue culture with Vero cells, positive samples to *T. gondii* could be detected [30].

### 2.3. Microscopic Methods

Oocysts, tachyzoites, or tissue cysts of *T. gondii* cannot be detected by gross inspection but can be identified by microscopy. In fact, microscopic methods are used for the detection of oocysts in fresh products and shellfish. Parasites are visible using nonspecific stains such as Giemsa or hematoxylin and eosin, but the use of specific stains with fluorescence-conjugated enzymes or antibodies allows them to be differentiated from other structures or apicomplexan parasites and increases the sensitivity [63]. The main disadvantage of microscopy as a detection technique is the appearance of false negatives. This is due to the small sample size that can be examined. In this way, not finding parasites in the examined sample is possible, even though there is contamination in other areas of the sample.

### 2.4. Molecular Methods

The polymerase chain reaction (PCR) is based on the in vitro amplification of specific DNA sequences. For these sequences, the DNA that is present in the analyzed samples is extracted, and several amplification cycles are carried out. The presence of parasite-specific DNA in the sample is visualized by agarose gel electrophoresis. If the sample contains the target DNA, a specific band is observed in the gel [35].

There are different targets that are available for the detection of *T. gondii* by PCR. The most common are the B1 fragment, which is repeated 35 times in the parasite genome, and a region of 529 bp that is repeated 200–300 times [64]. However, commercial DNA isolation methods are usually designed for 25 mg of sample, but tissue cysts are rare and, therefore, the chance of detecting *T. gondii* in such a small sample is low. To allow analysis of large samples and to increase the detection sensitivity, methods that are based on artificial digestion, homogenization and isolation on Percoll gradients, and sequence-based magnetic capture have been described [63]. This allows the sample to be concentrated and more tissue can be analyzed. In addition, this simulates the conditions of our body when digesting food. The sample is incubated for one hour at 27 ºC with hydrochloric acid, pepsin, and sodium chloride, which causes the rupture of the tissue cyst walls of *T. gondii* and the release of bradyzoites [65].

Quantitative PCR (qPCR) is a variant of conventional PCR, which allows the detection of the parasite DNA concentration in the analyzed sample with elevated sensitivity, precision, and speed than conventional PCR, in addition to not requiring the use of gels. For the detection of *T. gondii* by qPCR the most widely used fluorophores are SYBR Green and TaqMan probes [39,66]. The SYBR Green fluorophore has higher sensitivity, but is more likely to bind non-specifically, whereas TaqMan probes have high specificity, but less sensitivity and, therefore, cannot detect low concentrations of parasite DNA [67]. The PCR method has been improved by fine-tunning multiplex PCR for the detection of different organisms simultaneously. More recently, Temesgen et al. (2019) developed and evaluated a new multiplex qPCR for the simultaneous detection of different parasites, including *T. gondii*, in berry fruits [41]. The results showed that it is a highly specific, precise, and robust method, which has potential application in food analysis laboratories. Shapiro et al. (2019) developed a multiplex PCR for the simultaneous detection of parasites, including *T. gondii*, in spinach. This method was found to be more sensitive than traditional qPCR [19].

Loop-mediated isothermal amplification (LAMP) enables DNA amplification with high sensitivity and specificity, efficiency, and speed [68]. It is a technique that uses a DNA polymerase with chain displacement activity, with four to six primers that are designed to recognize six to eight different regions of the target DNA, which allows the amplification specificity of LAMP to be very high. Up to 109 copies can be amplified in less than one hour under isothermal conditions (63–65 °C) [69]. These conditions facilitate the process, so a simple incubator is sufficient to amplify the DNA, which allows the use of this technique under field conditions. DNA amplification can be detected by visual inspection of the turbidity or fluorescence of the sample, or by real-time turbidimeter [68]. Therefore, it does not require gel electrophoresis, which reduces the test time and allows this technique to be a fast and accurate molecular method for the detection of *T. gondii*. For the detection of oocysts in fresh products, an adaptation of the LAMP technique has been developed. This new technique includes a chromatographic detection system with a lateral flow test strip that allows to accelerate the visualization of the results [47]. In 2013, the LAMP technique with reverse transcriptase (RT-LAMP) was developed for the detection of *T. gondii* in meat samples. The results suggest that RT-LAMP is a simple and reliable tool to detect meats that are contaminated with *T. gondii* [70]. LAMP seems to be an alternative to most expensive molecular methods with similar sensitivity, with a low detection limit of five oocysts per gram of tissue, and five oocyst per milliliter of hemolymph in bivalves [46].

These molecular techniques detected the DNA of the parasite, and genotyping is possible with them. However, different available genotyping methodologies have been irregularly applied in different geographic areas and over different matrices [71]. The main drawback of these molecular techniques is that it only allows the DNA of the parasite to be detected, but the viability of *T. gondii* is unknown. So, other methods are required to establish whether the detected DNA belongs to viable parasites [31]. Until now, one molecular method for viability detection has been developed. Propidium monoazide-based qPCR (PMA-qPCR) has been positively evaluated [72], and its ability to detect viable parasites in leafy greens has been demonstrated recently [43].

### 2.5. Serological Methods

Serological methods are indirect methods that are intended to confirm infection with *T. gondii* in animals and humans, but they have been adapted for testing meat and meat juices. Generally, they serve as a first screening to detect seropositive animals, in which later the infection will be confirmed in the tissue samples through a bioassay. However, these methods can also be used to detect infection in meat juice samples, for example [73]. The serological methods that are used to detect antibodies against *T. gondii* in serum or meat juice are indirect hemagglutination antibody (IHA), the latex agglutination test (LAT), indirect fluorescent antibody test (IFAT), modified agglutination test (MAT), Western blot, and ELISA, with MAT, IFAT, and ELISA being the most used and validated methods [74]. All these techniques detect immunoglobulins (Ig) G and M in serum or tissue fluid. The MAT technique is more sensible than other agglutination methods, but it is not useful for slaughterhouse use, as it requires a large number of intact tachyzoites [56]. The ELISA technique has been shown to be more sensitive and efficient than MAT for the detection of antibodies to *T. gondii* [75,76]. The serological methods are quick and easy to perform, but they have certain limitations. The sensitivity and specificity can vary, and the results do not always correlate with bioassay results [73].

### 2.6. New Methods of Detection

The traditional methods of detection have limitations and there are no standardized protocols for their application in the food industry. For this reason, new detection methods are being developed for *T. gondii* that improve the efficiency and reproducibility.

In fresh products in particular, oocyst detection methods are scarce. In the last years, different authors have been developed methods for their detection. Lalonde and Gajadhar (2016) developed real-time PCR methods for the identification of protozoan oocysts in vegetables and fruits [76]. Slana et al. (2021) exhaustively described the different molecular methods for the detection of *T. gondii* in fresh products [77]. In bivalve mollusks, alternative detection molecular methods have been proven. Concretely, the Q3 lab-on-chip real time-PCR platform, a miniaturized platform, has been checked for the detection of *T. gondii* and other protozoan, with better results for *Toxoplasma* than other molecular approaches [78]. The DNA extraction using the bead-beating method has been demonstrated a rapid and simple method for detection in bivalves, but it is not valid for quantification [79]. The determination of *T. gondii* genotypes can provide relevant information for the control of this zoonosis. For this reason, different studies have evaluated detection and genotyping methods. Recently, similar sensitivity and specificity has been observed of the B1 and ROP8 genes for detection, whereas the latter seems more appropriate for genotyping [80].

One of the most relevant steps for molecular methods detection is the DNA extraction approach, that depends on matrix analyses. However, few studies have been done that are related it. Temesgen et al. (2020) compared two commercially available DNA extraction procedures in berry fruits [81]. On the other hand, Gisbert-Algaba et al. (2017) have developed a method for its use in meat based on DNA extraction by magnetic capture, which has proven to be sensitive, economical and reliable, and validated by ISO 17025 [82]. This technique is a potential alternative to the mouse bioassay for the screening of various types of tissue and meat, with the advantage of being quantitative. Now, this is the most validated method for the detection of *T. gondii* in food, but it requires further validation before it can be applied to other food samples. Furthermore, since qPCR only allows determining the presence of the parasite, but cannot directly confirm the viability [31], recently some authors are fine-tunning the analysis of *T. gondii* RNA using reverse transcription and subsequent PCR (RT-PCR). This technique uses the enzyme reverse transcriptase to synthetize complementary DNA (cDNA) from the RNA molecules that are present in the sample. Although this technique has a high sensitivity, RNA degrades much faster than DNA and it is more easily contaminated, so this technique must be performed by highly qualified and experienced personnel. It has difficulty detecting tissue cysts, since it needs the parasite to be metabolically active at the time of analysis [66]. Recently, the cloth-based hybridization array system (CHAS) has been developed to confirm of PCR-positive results as a cheaper and easier method than sequencing [36] and a new real-time isothermal amplification method (real-time recombinase-aided amplification, RT-RAA) with more sensitivity and specificity than traditional RT-PCR has been tried in pork blood samples [83].

Loreck et al. (2019) developed a protein microarray for the simultaneous detection of IgG antibodies against different zoonotic agents and pathogens that cause disease in pigs, among which is *T. gondii* [84]. This is an efficient and valid method for detection since it allows the detection of antibodies against these zoonotic agents in a single measurement. Duong et al. (2020) developed a Luciferase-linked antibody capture assay (LACA) to detect *T. gondii* in serum chicken, and they obtained high sensitivity (90.5%) and specificity (95.4%) [85]. The best results were obtained by Fabian et al. (2020) that recently developed a new serological method that was named bead-based multiple assay (BBMA) using the Luminex technology with high sensibility (98.5%) and specificity (100%) relative to a reference of ELISA, IFAT, and MAT [53].

Different recent studies are aimed at improving sensitivity, looking for alternatives to bioassays that allow detecting the viability of the parasite, and a possible validation to be able to apply in the food industry. Moreover, the detection of different parasites simultaneously is relevant to the food industry, so the lack of standardized protocols does not only refer to *T. gondii*, but which is generalized in all foodborne parasites, to a lesser or greater extent. In this context, a novel metabarcoding assay followed by next generation sequencing (NGS) has been developed to simultaneously detect *Cryptosporidium* spp., *Giardia* spp., and *T. gondii* in shellfish [86]. The application of this type of technique in other products would allow us to achieve the control of parasitic diseases that are transmitted by food. Furthermore, it would be interesting to have a method that allows the detection of all infectious stages of *T. gondii* (tachyzoites, tissue cysts, and oocysts). About this, Guggisberg et al. (2020) have fine-tuned a one-way sequential sieving method to identify different stages of parasites in lettuce [87]. Even so, much research is still required to be able to apply these methods in the food industry in the future and improve the current situation of parasitic diseases, including toxoplasmosis.

## 3. Prevalence of *Toxoplasma gondii* in Food Products

Transmission through food is the main system of transmission of *T. gondii* to humans [18]. Tissues cysts and tachyzoites are responsible for infection thought meat and milk, respectively [56], and sporulated oocysts can contaminate fresh products, shellfish, and water, and infect humans after consumption [88]. Next, we will try to delve into the transmission mechanisms depending on the type of food.

### 3.1. Meat and Meat Products

*T. gondii* infections have been reported in all meat production animals around the world, although the prevalence depends on the detection method that is used (Table 1). Tissue cysts of parasites in meat are an important source of human infection, due to the fact of that these animals are secondary hosts of the parasite, which can survive long periods of time in these asymptomatic animals, which will later become meat products.

Different techniques are available to detect its presence. The mouse bioassay and PCR are the most widely used direct detection methods, followed by microscopy and the cat bioassay [31]. On the other hand, the MAT, IFAT, and ELISA tests are the most widely used serological methods for the detection of *T. gondii* infection in cattle and meat products [73]. Table 2 shows the animal and sample that was contaminated, the country of contamination, the method that was used for detection, and prevalence that was found.

Beef cattle may contain *T. gondii* cysts in their tissues, and they may pose a risk if meat from infected animals is eaten raw or undercooked [34]. Tissue cysts are less resistant to environmental conditions than oocysts. Even so, they remain infectious in refrigerated carcasses (around one to four degrees) or in minced meat for three weeks, that is, while the meat is fit for human consumption [151]. *T. gondii* DNA has been found in cured bacon, raw or smoked sausages, ham, and minced meat [128]. Infections of parasites are more frequent in lamb and pork than in beef and chicken, with sheep meat representing the highest risk of infection in humans [152]. Opsteegh et al. (2016) confirmed that 1.6% of the bovines that were analyzed by bioassay were positive, which indicates the presence of viable tissue cysts and, therefore, represents a potential risk for consumers [31]. In Italy, the seroprevalence was 8.7% of cattle, lower than the 13.4% of seroprevalence in animals that were imported [105]. The last data are in accordance with the results that were observed in other countries, such as Iran [107]. In Poland, using the direct agglutination test (DAT) method, the seropositive samples were 13% [106], and in Brazil, the IFAT method detected 40.6% of positive blood samples in beef cattle that were slaughtered [153]. Molecular methods indicated high values for meat cattle samples, with a 19.3% rate of positive results in Tunisia [100].

Sheep and goats present *T. gondii* in their consumable organs [89]. Serological and molecular methods have demonstrated that around of 10–24% of liver, meat, and heart samples were positive for sheep and goat [89]. In Italy, the meat juices from 28.6% sheep 27.5% goat were positive by commercial ELISA [97]; in Lebanon, the data are high, with a seroprevalence of 42% and 34% in sheep and goats, respectively [92], and other countries show even higher seroprevalences [94,104]. However, studies with molecular methods indicated a low prevalence in sheep and goat meat in Iran, with 14.4% and 11.1%, respectively [97], whereas in Tunisia, the data were 33.3% for sheep and 32.5% for goat meat samples [100], and in Australia molecular methods detected 43% of positive lambs that were examined [102]. To decrease this rate of infection, some strategies have been carried out. For example, in Denmark, seroprevalence in organic herds has been studied, concluding that organic herds present a higher prevalence, therefore, risk mitigation strategies in processing plants could be alternatives to serological surveillance [110]. This elevated prevalence of *T. gondii* in organic herds is due to the high risk of being exposed and infected with environmental oocysts of parasites or from the ingestion of infected rodents [154,155].

Pork meat consumption has been estimated to cause 41% of foodborne toxoplasmosis cases in the USA [156]. In Brazil, some studies indicated that 6.5% of pig serum that was examined was positive of *T. gondii* by IFAT and MAT, and 69.2% of them presented positive PCR in meat [119], whereas other studies show with the IFAT method, a 77% rate of seropositive animals, and the parasitic DNA was found in 66.7% of tissue samples that were recovered [120]. In Cuba, the seroprevalence was 13.3% [114]. In Poland, using the DAT method, the authors found 11.9% of seropositive animals [157]. Molecular detection showed a prevalence of 6.7% in India [122]. The results that were observed by different ELISA and molecular procedures, IFAT, and MAT showed that the test and cut-off that were used influence the results that were obtained [114,158]. The production system seems to be influential as well with the higher prevalence found in extensive systems or organic farms than in intensive ones [113,116]. However, Gomez-Samblas et al. (2021) found only a 10% of *T. gondii* infection in Iberian sows that were raised as outdoor livestock [159]. These authors did not find infection in cured products, so a correct and thorough curing process could eliminate the presence of the parasite. These data may be higher when the meat is not subjected to industrial processes. In Romania, where backyard pigs are a common practice in rural areas, the seroprevalence was 46.8% and 36.4% of meat that was evaluated presented the DNA of the parasite [113].

In China, Japan, and the USA, recent studies indicated a 10–20% infection rate of quick-frozen chickens [85,130,160]. In India, the chicken tissue prevalence is 2.3% and the seroprevalence is around 6.5% [130], whereas in Brazil the seroprevalence is around 36% [128]. *T. gondii* infection may be accompanied by infection from other pathogens of chickens, as *Eimeria tenella* [161]. Not only broilers have *T. gondii* infection. Positive laying hens have been found in samples of serum and organs [162]. But *T. gondii* has been detected in other species for that are not common in meat production or meat consumption. For example, the overall seroprevalence of *T. gondii* in water buffaloes was around 8% in Romania [135]. In the Czech Republic, the presence of this protozoan has been demonstrated in feathered game and ostriches, with a 5.4% of prevalence in wild ducks, 3.4% in common pheasants, and 36% of ostriches by molecular methods [134,150], while white-tailed deer presented 36% of seroprevalence in USA [147]. In Canada, the presence of *T. gondii* in serum and organs of wolverines (*Gulo gulo*) has been detected by different methods [163]. The common quails presented a seroprevalence of 13.1% in China [136]. In this country, donkey meat and Tolai hares’ consumption is common in some provinces. In these meats, the prevalence of *T. gondii* DNA was 9.2% and 8.1%, respectively [136,137].

In the last years, several studies have indicated that wildlife can be a source of infection by *T. gondii* and a reservoir of the parasite. Wild ruminants have been analyzed in different European countries and the studies showed a high seroprevalence in roe deer (39.6%), fallow deer (37.1%), red deer (16.6%), Southern chamois (14%), mouflon (11.5%), Iberian wild goat (7.8%), Barbary sheep (5.6%) [148], and other hunting species (Table 1). In the USA, the most common wildlife species with antibodies of parasite are feral swine (9% of prevalence) and venison (36%) [138,147]. From retail outlets, Plaza et al. (2020) estimated the presence of antibodies in 5.3% of beef, 14.3% chicken, 16.5% lamb, 14.1% pork, and 16% in venison samples in Scotland [164]. Wild boar is presented as the most relevant wildlife species for risk infection of *T. gondii* around the world, with a seroprevalence of 76.9% in Brazil [143], around 40% in Italy [144,145], 14% in USA [138], 15% in China [139], and 9% in Finland [140]. Crotta et al. (2022) detected 49% of seroprevalence in wild boars in Italy, with high percentage of co-infections with hepatitis E virus [140]. From these data, the conclusion is drawn that the presence of *T. gondii* in meat for human consumption is high in Asia, Europe, and the USA, with its detection and control before sale being of vital importance. 

### 3.2. Milk and Dairy Products

Tachyzoites can be shed in the milk of acutely infected animals, so both raw milk and raw dairy products can pose a risk of infection for consumers [165] (Table 3). In fact, one of the factors that is related to infection in the USA is the ingestion of unpasteurized goat’s milk [166]. Different studies show the presence of *T. gondii* in milk samples from sheep, goat, camels, and donkeys [52,167,168,169,170], where the prevalence can reach up to 43–65% [159]. However, these data differ between production procedures, management, and techniques of detection, increasing with deficient biosecurity levels (related to the application of a health management program, vaccination protocols, correct quarantines, protocols for visitors, etc.) [171], and were higher for serological rather than molecular techniques. In goat, molecular techniques revealed the presence in 20.6% of milk samples, whereas the ELISA showed 63.3% [172]. Other studies showed that relationship between the prevalence of *T. gondii* antibodies in the goat serum with a prevalence of *T. gondii* DNA in milk samples [157].

Although the transmission of *T. gondii* through cow’s milk has not been detected [167], tachyzoite survival in milk pH conditions has been demonstrated [175], which could indicate that although they have not yet been detected, we could find tachyzoites in unpasteurized cow’s milk, making it a possible route of transmission. Milk is considered a potential source of infection since the infectious parasite in its tachyzoite form can be transmitted by animal fluids. The main detection methods that are used in raw milk samples include the detection of parasitic DNA by PCR-based tests, usually targeting the 529 bp repeat sequence [170], or the B1 gene [30]. However, the detection of *T. gondii* DNA does not allow the viability of the parasite to be determined. For this reason, other techniques have been used to determine the viability of parasites in milk and dairy products, including the viability assay in cell culture, where the cytopathic effect of tachyzoites on Hep-2 cells is measured, including mouse and cat bioassays [175]. Mouse and cat bioassays were used to detect *T. gondii* in the milk and cheese of goats, demonstrating that fresh milk and cheese are a source of transmission, so the protozoan survives cold-enzyme treatment [29]. The ELISA test has also been used to evaluate the presence of *T. gondii* antibodies in goat milk samples. The study showed that this technique in milk samples could easily be applied to detect the seroprevalence of *T. gondii*, although it does not allow the detection of tachyzoites [171].

### 3.3. Fresh Products and Vegetables

Fresh products can become contaminated with *T. gondii* oocysts from cat feces or contaminated water, and act as a source of infection in humans. Oocyst detection in environmental and food samples is difficult due to complications in separating and concentrating oocysts from complex matrices, such as raw vegetables, so there is a lack of optimized laboratory methods for its detection [41]. However, Dumètre and Dardé (2003) have proposed possible methods for the detection of *T. gondii* in water, soil, and food samples (mainly, fruit and vegetables), based on methods that are used for other protozoa [176]. Hohweyer et al. (2016) developed an immunomagnetic separation assay (IMS) targeting the cell wall of oocysts, although it is not yet commercially available [67]. In addition to conventional methods such as microscopy, PCR or qPCR, a LAMP test has been developed to detect *T. gondii* in experimental contaminated baby ready-to-eat lettuces. The detection limit of this method was approximately 25 oocysts per 50 g of lettuce leaves [47]. Recently, special RT-PCR assay has been developed and it was effective to discriminate viable *T. gondii*, detecting two to nine oocysts per gram of spinach [43].

The first detection of *T. gondii* DNA in fruits and vegetables was in 2012 [177]. Nowadays, some studies have linked acute outbreaks of human toxoplasmosis with the ingestion of oocysts, where green vegetables have been identified as a possible vehicle of infection which can be contaminated by irrigation water [178,179,180]. In fact, Pinto-Ferreira et al. (2019) undertook a meta-analysis and concluded that vegetables will be the most common possible route of transmission in the future [181]. Contamination by *T. gondii* has been observed in different vegetables around the word, including lettuce, chicory, rocket, parsley, spinach, pack choi, cabbage, rape, asparagus, endive, Chinese chives, carrots, cucumbers, strawberries, and radish [182] (Table 4).

In Poland, vegetables from shops and home gardens presented a contamination rate by *T. gondii* of 9.7% [178]. In China, the prevalence of DNA protozoan was detected in 3.6% in vegetable analyses [42], whereas in Morocco these data increases to 21.2% [187]. In Italy, other studies did not found *T. gondii* in fresh produce [188] or it was at a low prevalence (0.8%) [183]. The prevalence in packaged ready-to-eat mixed salads was investigated by microscopic examination and detection by PCR, and the results revealed that 0.8% of the ready-to-eat salads were positive for *T. gondii*, where a high oocyst burden was found (from 62 to 554 per gram of vegetable) [188]. Also with molecular and microscopic methods, a mean of oocyst concentration in salad has been detected of approximately 23.5 oocysts per gram [37].

### 3.4. Marine Products

Aquatic environments can be contaminated with wastewater carrying *T. gondii* oocysts. Mollusks such as clams, mussels, oysters, and scallops, filter-feed and trap phytoplankton in the gills. This filter feeding process can also concentrate waterborne pathogens within their tissues, including oocysts, which can survive for long periods of time in both fresh- and salt-water [55]. For detection in mollusks, samples of whole tissue or organs can be used and the most frequent techniques that are used are those that are based on PCR, generally directed to the B1 gene [189,190]. Various molecular methods have been used for detection in fish, such as PCR, qPCR, and RT-PCR, targeting the same gene, or the 529 bp DNA repeat element. The last method seems more sensitive, with the five oocysts as a low limit of detection. But it is no more specific, requiring direct sequencing for definitive confirmation of *T. gondii* [191]. In addition, the techniques have been carried out in different matrices, such as the digestive tract, muscle, brain, and even gills, among others [192]. Serological techniques have also been used for the detection in fish, such as ELISA, by detecting IgG and IgM, suggesting the fish are actually infected with *T. gondii* [193], rather than just serving as paratenic hosts such as shellfish.

The consumption of raw mollusks is considered a risk factor for *T. gondii* infection. Table 5 shows the prevalence of parasite in different mollusks, bivalves, and fishes.

Different studies showed the prevalence of infection in Mediterranean bivalves of 6.6% to 9.4% in countries such as Turkey and Italy [189,190,198]. In China, 2.8% of marine bivalve shellfish analyses were positive for the DNA of protozoan, and depended on the temperature and precipitation, with a higher presence of *T. gondii* with elevated temperatures and precipitations [194]. Similar results were found in New Zealand, where the prevalence was 16.4% [195]. Recently, the presence of *T. gondii* in fish has been investigated. There is still controversy about possible parasitic infection in cold-blooded hosts. Some studies support that these animals can act as mechanical vectors, such as mollusks, containing oocysts in their digestive system [194]. In fact, *T gondii* DNA has been found in different fish species of local fish markets [197] and marine animals species. A recent review showed high prevalence in mustelids (54.8%) and cetaceans (30.92%) [201].

## 4. Control and Food Safety

The control of *T. gondii* infection must be done at several levels. First, certain risk factors increase the prevalence of the parasite in farm animals. Hygienic management practices and correct management which involves keeping cats away from crops and gardens and animal feed, are essential to control this pathogen in farms [121]. Temperature and humidity control could decrease the survival and distribution of the parasite, as well as a late replacement of the animals, since older animals present higher prevalence than young ones [94,202]. The intensive systems of production present lower prevalence than extensive or semi-intensive ones [171]. In the same way, organic farms present higher prevalence than conventional farms, probably due to due to the high risk of being exposed and infected with environmental oocysts of parasites or from ingested infected rodents [110]. Nevertheless, the most important factor in all production systems seems to be the biosecurity level (control of exposition and infection of animals with environmental parasites and control of domestic animals that are infected near the farms) and early detection [110,112]. Consumption of fresh milk and dairy products are other of factors that cause *T. gondii* infection in humans. In fact, pasteurization of milk and milk products is also an important control measure. Undoubtedly, stopping consuming these types of products could considerably reduce the prevalence of infection in humans. On the other hand, as occurs in meat products, adequate hygienic and sanitary conditions on farms would lead to this reduction. In fresh products and vegetables, the most common mechanism of contamination is irrigation with water that is contaminated by oocysts, so sanitary control measures in irrigation water would be interesting. Furthermore, washing fresh produce after harvest and before consumption is an important control measure, since the chemical disinfectants are not effective [18].

The control of *T. gondii* in food production is essential. However, control measures during food inspection are not applied [21]. Currently, different methods of inactivation exist, although in the industry they are not applied directly for the control of this parasite. The most used methods of control are thermal methods, including both high and low temperatures. Heat treatments can destroy oocysts from both sporulated and non-sporulated strains. It is also possible to eliminate bradyzoites and tachyzoites, although the elimination of the first requires higher temperatures and longer times [58,203]. Relationship between raw meat or other animal products have been demonstrated by several studies. In meat products, the main control measure to prevent infection is an adequate cooking and proper prevention of cross-contamination [204]. In fact, *T. gondii* can be eliminated from meat in 5–6 min at 49 °C, in 44 s at 55 °C, or in 6 s at 61 °C [205]. Different meat products require different temperature conditions for inactivation. For example, beef should be cooked at least 63 °C; whereas pork meat, minced meat, and bushmeat at 71 °C; and poultry at 82 °C. In general, meat should be cooked to at least 67 °C before consumption. In dairy products, the pasteurization of milk, at 63 °C for 30 min is sufficient to eliminate tachyzoites [206]. Rani and Pradhan (2021) published an exhaustive study that was related to the survival of *T. gondii* during cooking and low temperature storage and concluded that the parasite was not found when the internal temperature reached 64 °C and below −18 °C [207].

However, these elevated temperatures are not applicable to all food matrices. This is the case of vegetables and fresh products [208]. Regarding inactivation by low temperatures, it has been shown that freezing can inactivate tissue cysts of *T. gondii*. To inactivate isolate tissue cysts, a minimum of three days is required at −20 ºC [209]. In addition to thermal methods, other non-thermal methods can be used for the inactivation, such as high-pressure processing [55,154,210], ionizing radiation [211,212], and curing or salt [34,75]. The inactivation of *T. gondii* in food for thermal and non-thermal methods has been extensively analyzed in the review that was published by Mirza et al. (2018) [213].

The inactivation of *T. gondii* in food products has been realized traditionally with high temperatures (thermal methods) and when cured and salted [207,214], whereas the non-thermal methods are presented as emerging technologies for the control of *T. gondii* in food. High pressure processing (HHP) is a novel method for liquid and solid food products where pressures of 340–550 MPa during 1 min can inactive cysts of the parasite [215]. The second new method is ionizing radiation (IR), which is capable of inactivating or killing *T. gondii* cysts in meat [58]. However, these methods have not yet been tested in other food matrices or to inactive other parasitic forms.

## 5. Future Perspectives

Currently, most of the control of *T. gondii* infection is carried out at home, setting recommendations on food consumption in the groups that are most vulnerable to the parasite. This situation occurs because there are no regulations governing control measures against *T. gondii* in the food industry. Inactivation methods have yet to be optimized and validated to be applied against this parasite is a systematic way. More, different prevention measures could also be applied to its control. In farms, biosecurity and control will be factors of great relevance for infection control. Other measures such as restricting the access of cats to crops, gardens, and livestock feed, or the development of a vaccine that is aimed at cats to prevent the active release of oocysts could be used. Prevention measures could also be implemented at the livestock farm, such as the vaccination of cattle. Today, a vaccine against *T. gondii* is available for sheep, which prevents the spread of parasites to the placenta, and is used for the prevention of abortions in this species [216]. This vaccine also prevents the spread to other tissues, reducing the development of tissue cysts [217]. This measure seems to be a promising strategy, but it is still in the experimental phase and needs further development.

It would be useful to carry out a follow-up program at slaughter, detecting the meat that is positive for *T. gondii* and deriving its use for preheated or frozen meat products since, as we have seen, these methods are effective for the inactivation, as well as marking negative products as free from *T. gondii* [154]. Detection methods could be improved, mainly molecular methods given their high sensitivity, so that they can differentiate viable and non-viable parasites or use more than one detection method simultaneously (serological and molecular, for example). However, regulatory testing in meat animals is generally not considered practical due to the high prevalence in meat animals, i.e., many animals or carcasses would be found positive and would need to be destroyed or used for pre-cooked products. In short, a set of preventive measures, detection methods, and fine-tuning of inactivation methods are required to achieve control of this parasite and produce safe food for consumers.

## 6. Conclusions

*T. gondii* is the food parasite with the greatest epidemiological relevance, which is distributed worldwide, and with a complex life cycle that makes its detection very difficult. The main foods that are involved in the transmission of this parasite are meat and fresh products (vegetables and fruits) products through tissue cysts, mollusks, and fish, as well as through oocysts, and milk and dairy products through tachyzoites. Currently, the main detection methods are bioassays, in vitro culture, molecular methods (PCR and LAMP), and microscopy as direct methods, and serological techniques as MAT, IFAT, and ELISA as indirect methods. Due to the limitations of these methods, the emerging detection methods are aimed at developing methods with greater sensitivity and reproducibility, and generally, are aimed at the detection of several parasites simultaneously, which would increase their efficiency and facilitate their application in the food industry. Control methods include thermal methods such as heat, cooking, and freezing, as well as non-thermal methods such as HPP, IR, curing, or salting. Most of the control of *T. gondii* is carried out at home since there are no microbiological criteria for this parasite in the food industry and, therefore, it is not mandatory to comply with control measures. In the future, new detection methods should be validated to optimize the control of infection in food and apply them in the food industry.

## Figures and Tables

**Figure 1 foods-11-02542-f001:**
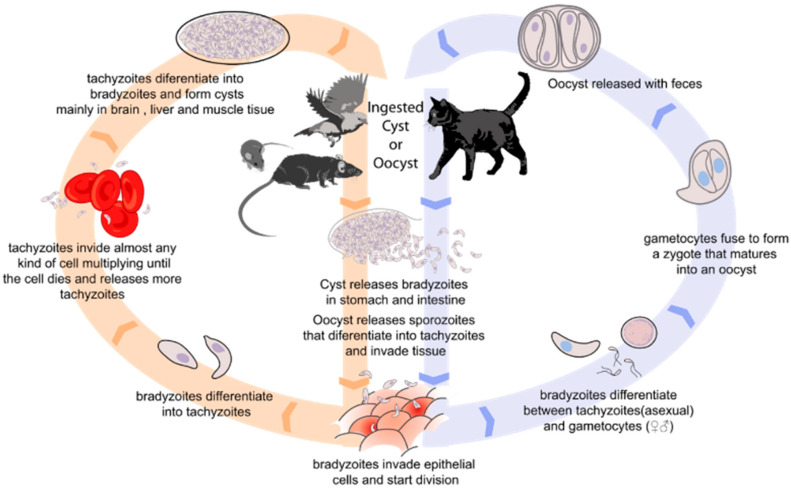
Biological cycle of *Toxoplasma gondii*.

**Table 1 foods-11-02542-t001:** The table shows different methods for *T. gondii* detection, sensitivity of method, and type of food product where this method has been used.

Detection Method	Specific Method ^1^	Type of Food Product	Detection Range (Sensitivity) ^2^	References
Animal model bioassay	Cat	Milk	25%	[29,30]
		Meat	100%	[31]
	Mouse	Milk	100%	[29]
		Meat	100% (10 tachyzoites)	[24]
		Fresh products	13%	[32]
		Bivalve mollusks	2.5%	[33]
		Water	100%	[34]
Cell culture		Meat	100% (10,000 tachyzoites)	[24]
		Milk	-	[30]
Microscopic method		Meat	-	[31]
Molecular methods	PCR	Meat	47.1%	[35]
		Fresh products	95–100%	[36,37]
		Water	100%	[36]
		Milk	100%	[29,38]
		Cheese	100%	[29]
	qPCR	Meat	92.3% (limit 0.01 pg)	[39,40]
		Fresh products	100% (1 oocyst)	[41,42,43]
		Bivalve mollusks	100%	[44]
		Water	100%	[44]
	LAMP	Lymph nodes	85.7%	[45]
		Mussels	5 oocyst/g	[46]
		Fresh products	25 oocyst/50 g	[47]
		Water	100% (1 fg)	[48,49]
Serological methods	IHA	Meat Juice	100% (10,000 oocysts)	[50]
	IFAT	Meat	97%	[51]
		Meat Juice	96.9% (10,000 oocysts)	[50]
	MAT	Meat	86.6%	[51]
		Milk	-	[52]
	ELISA	Milk	-	[30]
		Meat	91%	[51]
		Meat Juice	100% (10,000 oocysts)	[50]
	BBMA	Meat	98.5%	[53]

**^1^** PCR: Polymerase chain reaction; qPCR: real-time PCR; LAMP: Loop-mediated isothermal amplification; IHA: indirect hemagglutination antibody; IFAT: indirect fluorescent antibody test; MAT: modified agglutination test; ELISA: Enzyme-Linked Immunosorbent Assay; BBMA: bead-based multiplex assay. ^2^ The column shows the percentage of samples that were positively detected by the method and the quantity of parasites per quantity of food product that was detected if this data is known. The value (-) means that this data is not known.

**Table 2 foods-11-02542-t002:** *T. gondii* in animals and animal products. The table shows the producer animal, sample contaminated, country of contamination, method that was used for detection, and prevalence that was found.

Animal	Sample Analyzed	Detection Method ^1^	Number of Samples Tested	Number of Positive Samples (%)	Location	Reference
Sheep	Serum	ELISA	150	26 (17.3%)	Iran	[89]
	Serum	ELISA	550	59 (10.8%)	Iran	[90]
	Serum	ELISA	1039	179 (17.2%)	Latvia	[91]
	Serum	MAT	100	42 (42%)	Lebanon	[92]
	Serum	ELISA	64	30 (47%)	Slovakia	[93]
	Serum	DAT	252	148 (58.2%)	Ethiopia	[94]
	Liver	PCR	150	26 (17.3%)	Iran	[89]
	Liver	PCR	90	13 (14.4%)	Iran	[95]
	Heart	PCR	150	48 (32%)	Iran	[89]
	Brain and heart	MAT	136	10 (7.4%)	India	[96]
	Meat juice	ELISA	227	126 (28.6%)	Italy	[97]
	Meat juice	MAT	166	11 (6.6%)	China	[98]
	Meat	PCR	150	33 (22%)	Iran	[89]
	Meat	PCR	438	43 (9.8)	China	[99]
	Meat	PCR	150	50 (33.3)	Tunisia	[100]
	Meat	ELISA	109	38 (34.9%)	Malaysia	[101]
	Meat	PCR	79	34 (43%)	Australia	[102]
	Meat	PCR	177	3 (1.7%)	India	[103]
Goat	Serum	ELISA	150	16 (10.7%)	Iran	[89]
	Serum	ELISA	185	37 (20%)	Iran	[90]
	Serum	ELISA	445	189 (42.5%)	India	[104]
	Serum	MAT	80	27 (34%)	Lebanon	[93]
	Serum	ELISA	39	8 (21%)	Slovakia	[93]
	Serum	LAT	116	64 (55.2%)	Ethiopia	[94]
	Liver	PCR	150	24 (16%)	Iran	[89]
	Liver	PCR	90	8 (8.8%)	India	[95]
	Heart	PCR	150	36 (24%)	Iran	[89]
	Brain and heart	MAT	57	4 (7%)	India	[96]
	Meat juice	ELISA	51	14 (27.5%)	Italy	[97]
	Meat	PCR	150	26 (17.3%)	Iran	[89]
	Meat	PCR	254	27 (10.7)	China	[99]
	Meat	PCR	120	39 (32.5)	Tunisia	[100]
	Meat	ELISA	75	41 (54.7%)	Malaysia	[101]
	Meat	PCR	223	3 (1.3%)	India	[104]
Cattle	Serum	ELISA	57	13 (22.8%)	Italy	[105]
	Serum	DAT	2411	313 (13%)	Poland	[106]
	Serum	ELISA	400	52 (13%)	Iran	[107]
	Serum	IFAT	500	2.3 (40.6%)	Brazil	[108]
	Meat	PCR	150	29 (19.3)	Tunisia	[100]
	Meat	ELISA	392	98 (25%)	Malaysia	[101]
	Meat	PCR	48	5 (10.4%)	Brazil	[108]
Pig	Serum	ELISA	653	4 (0.6%)	Finland	[109]
	Serum	ELISA	447	73 (16.3%)	Denmark	[110]
	Serum	DAT	3111	370 (11.9%)	Poland	[106]
	Serum	IFAT	94	44 (46.8%)	Romania	[111]
	Serum	ELISA	420	56 (23.3%)	Cuba	[112]
	Serum	ELISA	370	14 (3.8%)	Italy	[113]
	Serum	ELISA and IFAT	127	56 (44.1%)	Italy	[114]
	Serum	MAT	375	8 (2.1%)	Italy	[115]
	Serum	ELISA	414	214 (51.7%)	Italy	[116]
	Serum	MAT	182	31 (17%)	Serbia	[117]
	Serum	MAT and IFAT	356	25 (7%) and 48 (13.5%), respectively	Brazil	[118]
	Serum	MAT and IFAT	400	26 (6.5%)	Brazil	[119]
	Serum	IFAT	60	44 (77%)	Brazil	[120]
	Serum	IHA	784	156 (19.9%)	China	[121]
	Tongue	PCR	60	20 (33.3%)	Brazil	[120]
	Tongue and muscle	PCR	810	54 (6.7%)	India	[122]
	Brain	PCR	339	34 (10%)	China	[123]
	Brain	PCR	107	51 (47.7%)	Italy	[116]
	Heart	PCR	94	25 (26.6%)	Romania	[111]
	Heart	qPCR	103	12 (11.6%)	Italy	[124]
	Diaphragm	PCR	45	15 (33.3%)	Serbia	[117]
	Diaphragm	PCR	1223	107 (8.7%)	China	[125]
	Diaphragm	PCR	60	24 (40%)	Brazil	[120]
	Diaphragm	qPCR	103	2 (1.9%)	Italy	[126]
	Tissue of seropositive animals	Mouse bioassay	26	18 (69.2%)	Brazil	[119]
	Muscle	PCR	60	23 (38.3%)	Brazil	[120]
	Meat juice	ELISA	212	33 (15.6%)	Denmark	[110]
	Meat	qPCR	118	46 (39%)	Brazil	[126]
	Meat	PCR	498	165 (33.1%)	Italy	[64]
	Meat	PCR	49	3 (6.1%)	Brazil	[108]
	Raw meat products	PCR	3223	175 (5.4%)	Poland	[127]
Chicken	Serum	IFAT	200	72 (36%)	Brazil	[128]
	Serum	ELISA	522	34 (6.5%)	India	[129]
	Serum	LACA	267	29 (10.9%)	Japan	[85]
	Brain	Mouse Bioassay	14	2 (14.3%)	Brazil	[128]
	Heart juice	MAT	1185	230 (19.4%)	USA	[130]
	Muscle and heart	PCR	522	12 (2.3%)	India	[129]
	Meat	PCR	257	21 (8.2%)	China	[131]
Ducks	Meat	PCR	115	9 (7.8%)	China	[131]
Geese	Meat	PCR	42	2 (4.8%)	China	[131]
Rabbit	Brain and heart	PCR	470	13 (2.8%)	China	[132]
Kibbeh	Meat	PCR	44	1 (2.3%)	Brazil	[108]
Water Buffalo	Serum	MAT and ELISA	197	16 (8.1%) and 13 (6.6%), respectively	Romania	[133]
Ostriches (farmed)	Serum	LAT	409	149 (36%)	Czech Republic	[134]
Common quails (farmed)	Serum	MAT	620	59 (9.5%)	China	[135]
Donkey (farmed)	Meat	PCR	618	57 (9.2%)	China	[136]
Tolai hares (farmed)	Serum	PCR	358	29 (8.1%)	China	[137]
	Brain	PCR	358	23 (6.4%)	China	[137]
Feral swine	Serum	ELISA	376	34 (9%)	USA	[138]
Wild boar (farmed)	Serum	LAT	882	88 (10%)	China	[139]
Wild boar	Serum	ELISA	331	164 (49%)	Italy	[140]
	Serum	ELISA	181	17 (9%)	Finland	[141]
	Serum	IFAT	26	20 (76.9%)	Brazil	[142]
	Serum	ELISA	306	61 (20%)	Germany	[143]
	Tissue	Mouse bioassay	22	1 (4.5%)	Brazil	[142]
	Brain	qPCR	141	44 (31.2%)	Italy	[144]
	Brain	PCR	263	58 (22%)	Italy	[145]
	Heart	qPCR	166	47 (28.3%)	Italy	[144]
	Heart	PCR	310	70 (22.6%)	Italy	[145]
	Muscle	qPCR	165	40 (24.2%)	Italy	[144]
	Muscle	PCR	311	74 (23.8%)	Italy	[145]
	Meat juice	ELISA	97	42 (43.3%)	Italy	[146]
	Meat	qPCR	306	37 (12%)	Germany	[143]
Venison	Serum	MAT	914	329 (36%)	USA	[147]
	Heart	Mouse bioassay	36	11 (30.6%)	USA	[147]
Roe deer	Serum	LAT	356	141 (39.6%)	Spain	[148]
	Serum	ELISA	323	130 (40.2%)	Italy	[149]
	Serum	ELISA	184	20 (11%)	Germany	[143]
	Meat	qPCR	184	11 (6%)	Germany	[143]
Fallow deer	Serum	LAT	372	138 (37.1%)	Spain	[150]
	Serum	ELISA	167	17 (10%)	Slovakia	[93]
	Meat	qPCR	80	2 (2%)	Germany	[143]
Red deer	Serum	LAT	553	92 (16.6%)	Spain	[148]
	Serum	ELISA	96	19 (19.8%)	Italy	[140]
	Serum	ELISA	65	4 (6%)	Germany	[143]
	Meat	qPCR	65	2 (2%)	Germany	[143]
Southern chamois	Serum	LAT	186	26 (14%)	Spain	[148]
Mouflon	Serum	LAT	209	24 (11.5%)	Spain	[148]
	Serum	ELISA	50	12 (24%)	Italy	[140]
Iberian wild goat	Serum	LAT	346	27 (7.8%)	Spain	[148]
Chamois	Serum	ELISA	104	4 (3.8%)	Italy	[140]
Barbary sheep	Serum	LAT	18	1 (5.6%)	Spain	[148]
Moose	Serum	DAT	463	111 (23.9%)	Estonia	[149]
Wild ducks	Brain	qPCR	280	7 (2.5%)	Czech Republic	[150]
	Heart	qPCR	280	11 (3.9%)	Czech Republic	[150]
	Muscle	qPCR	280	4 (1.4%)	Czech Republic	[150]
Common pheasants	Brain	qPCR	350	8 (2.3%)	Czech Republic	[150]
	Heart	qPCR	350	4 (1.1%)	Czech Republic	[150]
	Muscle	qPCR	350	3 (0.9%)	Czech Republic	[150]

**^1^** ELISA: Enzyme-Linked Immunosorbent Assay; MAT: modified agglutination test; DAT: direct agglutination test; PCR: Polymerase chain reaction; LAT: latex agglutination test; IFAT: indirect fluorescent antibody test; qPCR: real-time PCR.

**Table 3 foods-11-02542-t003:** *T. gondii* in milk and dairy products. The table shows the producer animal, sample contaminated, country of contamination, method that was used for detection, and prevalence that was found.

Animal	Sample Analyzed	Detection Method ^1^	Number of Samples Tested	Number of Positive Samples (%)	Location	Reference
Donkey	Milk	ELISA	418	41 (9.2%)	China	[167]
Goat	Milk	ELISA	30	19 (63.3%)	Italy	[172]
	Milk	PCR	60	39 (65%)	Poland	[157]
	Milk	ELISA and qPCR	30	27 (90%) and 1 (3.3%), respectively	Egypt	[173]
	Bulk tank milk	ELISA	100	59 (59%)	Italy	[172]
Sheep	Milk	PCR	58	1 (1.7%)	Mongolia	[168]
	Milk	ELISA and qPCR	30	18 (60%) and 1 (3.3%), respectively	Egypt	[173]
Camel	Milk	PCR	9	8 (88.9%)	Mongolia	[168]
	Milk	ELISA and qPCR	30	1 (3.33%) and 0 (0%), respectively	Egypt	[173]
Cattle	Bulk tank milk	ELISA	149	8 (5.4%)	Iran	[174]

**^1^** ELISA: Enzyme-Linked Immunosorbent Assay; PCR: Polymerase chain reaction; qPCR: real-time PCR.

**Table 4 foods-11-02542-t004:** *T. gondii* in fresh products and vegetables. The table shows the product that was analyzed, country of contamination, method used for detection, and prevalence that was found.

Product Analyzed	Detection Method ^1^	Number of Samples Tested	Number of Positive Samples (%)	Location	Reference
Mixed-salad packages	qPCR	648 packages	5 (0.8%)	Italy	[183]
	PCR	90 packages	8 (8.9%)	Czech Republic	[184]
Leafy greens	qPCR	152	45 (29.6%)	Morocco	[185]
Carrot	qPCR	30	3 (10%)	Morocco	[186]
	qPCR	46	9 (19.5%)	Poland	[177]
	PCR	93	7 (7.5%)	Czech Republic	[184]
Chicory	PCR	40	2 (5%)	Brazil	[187]
Red cabbage	qPCR	8	1 (1.2%)	China	[42]
Coriander	qPCR	29	8 (27.6%)	Morocco	[186]
Cucumber	PCR	109	13 (11.9%)	Czech Republic	[184]
Lettuce	qPCR	28	3 (10.7%)	Morocco	[186]
	qPCR	50	9 (18%)	Poland	[177]
	qPCR	71	5 (7%)	China	[42]
	PCR	168	5 (3%)	Brazil	[187]
Spinach	qPCR	50	2 (4%)	China	[42]
Parsley	qPCR	29	13 (44.8%)	Morocco	[186]
	PCR	5	1 (20%)	Brazil	[187]
Pak Choi	qPCR	34	1 (2.9%)	China	[42]
Radish	qPCR	16	1 (6.3%)	Morocco	[186]
	qPCR	60	3 (5%)	Poland	[42]
Rape	qPCR	22	1 (4.5%)	China	[42]
Rocket	PCR	7	1 (14.3%)	Brazil	[187]

**^1^** PCR: Polymerase chain reaction; qPCR: real-time PCR.

**Table 5 foods-11-02542-t005:** *T. gondii* in marine products. The table shows the animal, sample contaminated, country of contamination, method that was used for detection, and prevalence that was found.

Animal	Sample Analyzed	Detection Method ^1^	Number of Samples Tested	Number of Positive Samples (%)	Location	Reference
Bivalve shellfish	Tissue	PCR	2907	82 (2.8%)	China	[194]
Green-lipped mussels	Tissue	PCR	104	13 (16.4%)	New Zealand	[195]
Mediterranean mussel	Gills	qPCR	53 pools at 795 specimens	21 (39.6%)	Turkey	[189]
Clam	Tissue	qPCR	61 pools at 1020 specimens	4 (6.6%)	Tunisia	[190]
	Digestive gland	PCR	390	6 (1.5%)	Canada	[196]
	Haemolymph	PCR	390	2 (0.6%)	Canada	[196]
Mediterranean scald fish	Gills	PCR	1 pool at 6 specimens	1 (100%)	Italy	[197]
Pacific oyster	Gills	PCR	6 pools at 109 specimens	1 (16.67%)	Italy	[198]
Oyster	Mantle, gills, and rectum	qPCR	1440	447 (31%)	USA	[199]
Bogue	Gills	PCR	26 pools at 260 specimens	4 (15.4%)	Italy	[197]
	Intestine	PCR	26 pools at 260 specimens	3 (11.5%)	Italy	[197]
	Muscle	PCR	26 pools at 260 fish	6 (23.1%)	Italy	[197]
White seabream	Muscle	PCR	3 pools of 18 specimens	1 (33.3%)	Italy	[197]
European anchovy	Gills	PCR	35 pools at 350 specimens	2 (5.7%)	Italy	[197]
	Intestine	PCR	35 pools at 350 specimens	1 (2.9%)	Italy	[197]
European hake	Gills	PCR	15 pools at 90 specimens	1 (6.7%)	Italy	[197]
	Muscle	PCR	15 pools at 90 specimens	1 (6.7%)	Italy	[197]
Red mullet	Intestine	PCR	11 pools at 110 specimens	3 (27.3%)	Italy	[197]
American prawn	Muscle	PCR	618	4	China	[197]
Nippon shrimp	Muscle	PCR	813	1	China	[200]
Axillary seabream	Gills	PCR	8 pools at 80 specimens	2 (25%)	Italy	[197]
	Intestine	PCR	8 pools at 80 specimens	1 (12.5%)	Italy	[197]
	Muscle	PCR	8 pools at 80 specimens	1 (12.5%)	Italy	[197]
Common pandora	Gills	PCR	3 pools at 18 specimens	1 (33.3%)	Italy	[197]
	Intestine	PCR	3 pools at 18 specimens	2 (66.7%)	Italy	[197]
	Muscle	PCR	3 pools at 18 specimens	1 (33.3%)	Italy	[197]
Thornback ray	Muscle	PCR	1 fish	1 (100%)	Italy	[198]
Red scorpionfish	Intestine	PCR	1 pool at 3 specimens	1 (100%)	Italy	[197]
Blotched picarel	Muscle	PCR	4 pools at 24 specimens	1 (25%)	Italy	[197]
Atlantic horse mackerel	Muscle	PCR	15 pools at 120 specimens	4 (26.7%)	Italy	[197]

**^1^** PCR: Polymerase chain reaction; qPCR: real-time PCR.

## Data Availability

Not applicable.

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
