# Peer review of "Toxoplasma gondii* in Foods: Prevalence, Control, and Safety"

_foods, 2022, doi:10.3390/foods11162542_

Round 1

Reviewer 1 Report

I reviewed the manuscript entitled, Toxoplasma gondii in foods: prevalence, control, and safety. The manuscript is well written and contributes to the field. However, authors should add more emerging methods and technologies for detection and reduction of T gondii in food products as below.

Abstract is well written and appropriate

Introduction

Figure 1 quality should be improved. It is difficult to read small text

Introduction should be improved with latest studies while highlighting the research gap and need of conducting this study.

Section 2. different methods should be provided in table form. Also, highlight the detection range and type food products.

Lines 296-298: different font size and sentence is not ended with full stop

Tables 1-4. method names should be abbreviated as footnotes

Section 4: authors must add traditional and emerging technologies used to reduce the Toxoplasma gondii in foods

References are not according to the journal format. Foods uses journal abbreviation 

Author Response

Thank you very much for your comments and recommendation. The responses point to point are in the attached file.

Reviewer 2 Report

The manuscript entitled Toxoplasma gondii in foods: prevalence, control, and safety had analyzed the current detection methods, the prevalence of Toxoplasma gondii in different food products, and put forward the control measures. The relevant work has been studied, so it is not very innovative. But this manuscript had summarized the current research results to a certain extent, so that it has reference significance for relevant researchers.

Some specific concerns and suggestions are as follows:

1. The symptoms of Toxoplasma gondii infection in humans should be stated detailedly.

2. Section 1: The introduction is insufficient.

3. Lines 48-56: Here, the author introduces the infection route of Toxoplasma gondii, and whether there have been large-scale human infections of Toxoplasma gondii in the past. If yes, please list.

4. Section 2.1: Is there a big difference in the detection results between the cat model and the mouse model? Why does the mouse model detect cerebrospinal fluid and abdominal fluid instead of feces?

5. Section 2.6: The multiplex PCR method can be attributed to the molecular method as an improved method.

6. Section 2: All detection methods are based on DNA with own defects. Is there exist detection methods for live Toxoplasma gondii?

7. Section 3: Why meat is the main source of infection and what are the main reasons?

8. Section 5: The author should give a perspective on how to detect Toxoplasma gondii efficiently and sensitively in the future.

Author Response

(The authors gave the same response as above.)

Round 2

Reviewer 1 Report

Authors thoroughly answered the suggestions made by me. In my opinion, this version of the manuscript can be accepted for publication. 

Reviewer 2 Report

My concerns have been addressed in the resubmission.